# OpenReview forum: "Synthesizing Post-Training Data for LLMs through Multi-Agent Simulation"
_ICLR.cc/2025/Conference — Submitted to ICLR 2025_

### Official Review · Reviewer_pgvU · 2024-11-03

**Soundness:** 2
**Presentation:** 3
**Contribution:** 3
**Rating:** 5
**Confidence:** 4

**Summary:**

MATRIX is a multi-agent simulator that creates realistic, scalable scenarios. It features 1000 agents and a structured communication mechanism, inspired by human social homophily, to reduce meaningless interactions. These components enable MATRIX to generate diverse and realistic text-based scenarios.

**Strengths:**

1. The authors' motivation is clear.
2. The experimental section of the paper is quite detailed.
3. The methods section is thoroughly explained.

**Weaknesses:**

1. As far as I know, generating data through multi-agent interaction based on role-playing under predefined scenarios is widely used. Although the authors mention some distinctions from related works in the paper, I believe they need to further elaborate on the novelty of their work, particularly in comparison to other role-playing-based data generation or multi-agent interaction methods, such as AutoGen[1] .
2. The experiments in this paper are conducted only using the LLaMA-3-8B as the base model. It would be beneficial to evaluate the method on other models of varying sizes to demonstrate its robustness.
3. I believe the authors need to elaborate on the computational cost of the MATRIX framework.
4. I noticed that in Figure 4(a), when the number of agents increases from 10 to 100, there seems to be no improvement in performance on Arena-Hard. I believe this phenomenon is worth further investigation.

References:
[1] : https://github.com/microsoft/autogen

**Questions:**

Please refer to the above.

---

### Official Review · Reviewer_im5V · 2024-11-03

**Soundness:** 3
**Presentation:** 3
**Contribution:** 3
**Rating:** 6
**Confidence:** 4

**Summary:**

This work introduces a multi-agent simulation framework for automatically generating post-training data. The simulator is grounded in realistic agent profiles and produces diverse social scenarios with a structured communication mechanism to generate synthetic data. Remarkably, by generating only 20,000 instruction-response pairs, a model fine-tuned with this data can outperform Meta’s Llama-3-8B-Instruct model, which was trained on over 10 million pairs, in benchmarks such as Alpaca-Eval and Alpaca-Hard. Furthermore, the synthetic data generation process demonstrates strong control capabilities, enabling targeted augmentation of specific model aspects.

**Strengths:**

(1) Comprehensive Experiments: This work features extensive experiments and baseline comparisons, including evaluations against models like MagPie and WildChat on datasets such as UltraFeedback and Orca. Beyond general instruction tuning performance on benchmarks like AlphacaEval2 and Alpaca-Hard, it also compares results across domain-specific tasks like multi-turn dialogue, coding, and safety. These evaluations demonstrate MATRIX’s robust capability and data efficiency across diverse domains.

(2) In-Depth Analysis: In addition to experiments covering various aspects of general instruction tuning, this work provides an in-depth analysis of model scaling and scenario diversity in relation to performance. It also examines the quality and complexity of the generated data.

**Weaknesses:**

(1) The instruction tuning process of Llama-3-8B-Instruct encompasses a broader range of capabilities compared to AlpacaEval, including multilingual abilities, self-recognition of identity, and more nuanced safety settings. Consequently, it’s not fair to directly compare over 10 million instruction-tuning data points with 20,000 data points mentioned in the paper.

(2) Additionally, AlpacaEval itself compares model responses using GPT-4-0314 as a reference. However, OpenAI models are also used to generate instructions (although the specific model isn’t mentioned in the paper, only an OpenAI icon appears in the figure, more technical details about the generation process can be added). This could bias the results toward responses that align with GPT-4’s generation, making improvements on AlpacaEval and AlpacaHard potentially less solid. More human evaluations and case study could be helpful and beneficial to confirm that the improvements are indeed significant. Nonetheless, the results on domain-specific tasks show clear gains.

(3) Additional experiments are needed to check for data leakage, ensuring that generated instruction data does not have a high similarity score with the test data (e.g., avoiding overlap with problems in HumanEval or MBPP). This would verify that the synthetic data generation process does not inadvertently introduce data leakage.

**Questions:**

(1) I’m curious whether certain instruction data are especially important for driving improvement. Does each data point contribute equally to the training process, or are some instructions more influential?

(2) Based on Table 7, it seems that structured communication is the primary factor behind the method’s effectiveness. Could you provide further analysis on this? Like the quality and difficulty analysis on random communication and without communication results are pretty bad or so? A case study on why the communication mechanism leads to such substantial improvement would also be valuable.

(3) Regarding code instruction tuning data, is there any metric to demonstrate the quality gap between MATRIX-Code and OpenCodeInterpreter? I am wondering the reason why the results for OpenCodeInterpreter are so low? According to the OpenCodeInterpreter paper, the 7B CodeLlama model achieves 75.6 on HumanEval and 69.9 on MBPP. Additional technical details on the fine-tuning methods used would help clarify whether the comparison is fair.

---

### Official Review · Reviewer_PZzj · 2024-11-11

**Soundness:** 2
**Presentation:** 2
**Contribution:** 2
**Rating:** 3
**Confidence:** 4

**Summary:**

This paper introduces a synthetic data generation strategy for creating high quality SFT and DPO datasets. The proposed strategy uses a multi-agent simulator that explicitly incorporates real-world user requirements into the data synthesis process. This simulator generates realistic and diverse scenarios using 1000 agents and a structured communication mechanism, where each agent is built based on a real-world human profile. These scenarios are then combined with specific user requirement categories (such as coding, dialogue, safety, etc.) to generate high quality instruction data that captures a wide range of real-world human needs.

Experiments are conducted by training Llama3-8B model using synthetic datasets generated by the proposed approach and comparing it with Llama3-8B models trained using various alternative datasets. The results clearly demonstrate the effectiveness of the synthetic datasets generated using the proposed approach.

**Strengths:**

The overall idea of creating a multi-agent simulator based on real-world human profiles is interesting. Grounding data synthesis in real-world human behavior can result in more realistic data. Specifically, the proposed approach goes beyond recent PersonaHub and leverages interactions between agents to create complex and diverse scenarios.

Experimental results show that the synthetic datasets generated by the proposed approach are more effective than various existing real and synthetic SFT/DPO datasets.

**Weaknesses:**

Presentation: The main problem with this paper is lack of specifics to fully understand and replicate the approach. The papers describes the components of the proposed multi-agent simulator at a high level without providing concrete details.

The proposed approach uses 1000 agents created based on real-world human profiles. Many things are unclear from the paper:
-- What kind of user profiles are used in the simulator? What is the distribution? Paper does not provide details about these profiles.
-- Which web sources are used to gather data for these user profiles?
-- What kind of person-specific data is obtained from the web to create a user profile?
-- How is this web data processed by the LLM to create a user profile? What are the LLM prompts used?
-- Line 200 says "Each profile includes a unique anonymized name, description, and a record of past actions, all processed to protect privacy." What does an action mean here? What kind of actions of real humans are obtained from internet?

What are the LLM prompts used to create goals and action plans for each agent? Does each agent have multiple life goals?

Line 154-155: "For MATRIX-Gen-SFT, the instructions are generated with both simplicity and diversity. For MATRIX-Gen-DPO, the instructions are complex and specialized."
There are no details provided regarding the difference between "simple SFT instructions" and "Complex/specialized DPO instructions" and how the simplicity/complexity is varied. What types of instructions are considered simple and what are considered complex/specialized?

It was difficult to understand how the agents and modulators are operating/communicating. I couldn't follow what actions are being communicated and how the modulators actually operate. The paper neither provides concrete specifics nor examples describing the scenario generation mechanism.

Experimental evaluation:
According to Table 9 in Appendix, using specialized instructions (type 2) for SFT and simple instructions (Type 1) for DPO gives the best results (see the last row of Table. 9). However, the authors derive opposite conclusions and perform SFT with simple instructions and DPO with specialized instructions (as specified in lines 154-155).

It is unclear why authors chose different models as starting points for different domains when evaluating domain-specific Matrix-Gen SFT datasets (coding, safety and multi-turn dialogue).

The target responses for the synthesized instructions are generated using Llama3-8B-Instruct. Training with these responses can be interpreted as knowledge distillation. Despite distilling from Llama3-8B-Instruct, the Matrix-Tuned-Model outperforms Llama3-8B-Instruct (Table. 2 & 3). There is no discussion regarding this.

**Questions:**

Please see weaknesses section for questions regarding the proposed method and experimental results.

The biggest concern I have is lack of necessary details about the proposed approach. In the current state, the paper feels more like a technical report rather than a scientific paper that others from the community could reproduce and build upon.

**Details Of Ethics Concerns:**

In the rebuttal phase, authors have indicated that the approach presented in the paper heavily relies on user data crawled from Twitter. Authors mentioned that they used LLMs to remove personal information from this data. This was done by simply prompting as LLM using the following prompt:
***Given the user profile, please identify and remove any personal information such as names, ages, locations, or other identifiers from the following text. {User profile including their tweets}***

It is unclear how effective this processed is and how much of private user information is being used by the proposed approach.

---

> ### Comment · Reviewer_PZzj · 2024-11-25
> **Thank you for the response.**
>
> I thank the authors for a detailed response. While the response provides several additional details about how the agents are created, it is  still a bit difficult to clearly understand the agent communication mechanism. I am not sure if anyone could actually replicate the proposed synthesis strategy with the information provided in the current paper, especially given that everything is built based on user data crawled from Twitter.
>
> So, I still feel that this is more like a technical report rather than a scientific paper that others from the community could reproduce and build upon.

---

### Meta-Review · Area_Chair_ihgn · 2024-12-16

**Metareview:**

This paper proposes a multi-agent simulator framework for generating synthetic training data for language models. The key claim is that with only 20,000 instruction-response pairs generated through structured agent interactions, models fine-tuned on this data can outperform those trained on much larger datasets. The main strengths are comprehensive experiments across multiple tasks and detailed ablation studies. However, major weaknesses include: 1) Lack of technical details needed for reproducibility - the paper reads more like a technical report than a scientific paper, especially regarding the agent communication mechanisms and Twitter data processing, 2) Privacy concerns around using crawled Twitter data despite attempts at anonymization, 3) Questionable fairness in comparing 20K synthetic samples against Llama3's 10M instruction tuning samples given the latter's broader capabilities, and 4) Potential evaluation bias since AlpacaEval uses GPT-4 as reference. Due to these issues, particularly the reproducibility concerns and privacy risks, I recommend rejection.

**Additional Comments On Reviewer Discussion:**

The author response provided additional details on agent profiles, communication mechanisms, and data processing. However, one reviewer remained unconvinced about the replicability of the approach, noting it still lacks sufficient technical specifics especially given the reliance on Twitter data. Another reviewer was satisfied with the clarifications and raised their score. The third reviewer maintained concerns about novelty compared to existing multi-agent frameworks despite the authors' explanation of MATRIX's unique features. On privacy, the authors explained using LLM-based anonymization but didn't fully address the fundamental issues with crawling user data. The additional experiments on different model sizes and data leakage analysis were helpful but don't overcome the core reproducibility and privacy concerns. Weighing the mixed reviewer reactions and considering the serious reproducibility limitations, I maintain the rejection recommendation.

---

### Decision · Program_Chairs · 2025-01-22

Reject